# Untargeted Metabolome Analysis Reveals Reductions in Maternal Hepatic Glucose and Amino Acid Content That Correlate with Fetal Organ Weights in a Mouse Model of Fetal Alcohol Spectrum Disorders

**DOI:** 10.3390/nu14051096

**Published:** 2022-03-05

**Authors:** Nipun Saini, Manjot S. Virdee, Kaylee K. Helfrich, Sze Ting Cecilia Kwan, Sandra M. Mooney, Susan M. Smith

**Affiliations:** 1UNC Nutrition Research Institute, University of North Carolina at Chapel Hill, Kannapolis, NC 28081, USA; manjot_virdee@unc.edu (M.S.V.); kaylee.helfrich@gmail.com (K.K.H.); sk2563@cornell.edu (S.T.C.K.); sandra_mooney@unc.edu (S.M.M.); 2Department of Nutrition, University of North Carolina at Chapel Hill, Kannapolis, NC 28081, USA

**Keywords:** pregnancy, prenatal alcohol exposure, hepatic metabolism, untargeted metabolomics, glucose, amino acids, maternal–fetal metabolism, fetal weight, gluconeogenesis

## Abstract

Prenatal alcohol exposure (PAE) causes fetal growth restrictions. A major driver of fetal growth deficits is maternal metabolic disruption; this is under-investigated following PAE. Untargeted metabolomics on the dam and fetus exposed to alcohol (ALC) revealed that the hepatic metabolome of ALC and control (CON) dams were distinct, whereas that of ALC and CON fetuses were similar. Alcohol reduced maternal hepatic glucose content and enriched essential amino acid (AA) catabolites, N-acetylated AA products, urea content, and free fatty acids. These alterations suggest an attempt to minimize the glucose gap by increasing gluconeogenesis using AA and glycerol. In contrast, ALC fetuses had unchanged glucose and AA levels, suggesting an adequate draw of maternal nutrients, despite intensified stress on ALC dams. Maternal metabolites including glycolytic intermediates, AA catabolites, urea, and one-carbon-related metabolites correlated with fetal liver and brain weights, whereas lipid metabolites correlated with fetal body weight, indicating they may be drivers of fetal weight outcomes. Together, these data suggest that ALC alters maternal hepatic metabolic activity to limit glucose availability, thereby switching to alternate energy sources to meet the high-energy demands of pregnancy. Their correlation with fetal phenotypic outcomes indicates the influence of maternal metabolism on fetal growth and development.

## 1. Introduction

Fetal alcohol spectrum disorders (FASD) is an umbrella term used to describe the wide range of birth defects caused by prenatal alcohol exposure (PAE). These defects range from mild to severe forms of growth restriction, facial anomalies, and neurodevelopmental deficits leading to lifelong physical, metabolic, behavioral, and cognitive disabilities. Maternal nutrition is an important modifier of fetal alcohol outcome together with dose, frequency, and timing of alcohol exposure. Alcohol is known to reduce the availability of many maternal-derived nutrients essential for fetal growth [1,2]. We also know that selected individual nutrients, such as choline [3,4], iron [5], folate [6,7], and essential fatty acids (docosahexaenoic acid (DHA) and mixtures of omega-3 polyunsaturated fatty acids) [8,9,10,11], can improve FASD-associated behavioral and growth outcomes when given during pregnancy. However, little is known about other nutrient-PAE interactions, how PAE disrupts maternal nutrient utilization, and how such metabolic disruptions may contribute to fetal impairments.

Untargeted metabolomics analysis provides a comprehensive platform for biomarker discovery and to interrogate how metabolic pathways both drive and respond to disease states, information that informs the design of functional interventions [12,13]. To this point, metabolomics studies in FASD have focused on biomarker analysis and have identified ethyl glucuronide, phosphatidylethanol, fatty acid ethyl esters, and gamma-glutamyltransferase as biomarkers of exposure [14,15]. Most of these studies use biospecimens that can be readily assayed such as serum, plasma, urine, meconium, amniotic fluid, and cord blood. These biomarker studies are important for diagnostic applications; however, they may not inform mechanism. Plasma and urinary metabolite profiles reflect the analyte contributions from multiple peripheral organs, whereas other biospecimens are tightly regulated, factors that can complicate mechanistic understanding. To this end, we focus on the liver for metabolite analysis, to enhance the mechanistic understanding of maternal–fetal metabolism in FASD.

Liver is a central regulator of circulating macronutrients and their utilization [16,17]. Untargeted hepatic metabolomics has informed the mechanistic understanding of disease states, such as hepatocellular carcinoma [18], Alzheimer’s disease [19], liver injury [20], and non-alcoholic-fatty liver disease [21]. Liver is also a major site for alcohol metabolism, and the effect of alcohol in altering hepatic metabolic activity and macronutrient utilization is well known in non-pregnant alcoholics [22,23,24,25]. Chronic alcohol consumption generates reactive oxygen species, reduces antioxidant capacity, and enhances inflammation and steatosis; this latter is characterized by the accumulation of triglycerides, phospholipids, and cholesterol esters, and a reduction in fatty acid oxidation [26]. This is accompanied by reduced absorption and utilization of micronutrients, including vitamins A, B1 (thiamine), B2 (riboflavin), B6 (pyridoxine), C, D, E, and K, as well as folate, calcium, magnesium, phosphate, iron, and the trace elements zinc and selenium [22,27]. In contrast, little is known about how alcohol alters nutrient utilization during pregnancy, which is characterized by an exceptional anabolic demand in both mother and fetus. To gain mechanistic insights into the effects of alcohol in pregnancy, we performed untargeted metabolite analysis on the late-term maternal and fetal livers in a C57BL/6J mouse model of FASD. This model causes a modest decline in fetal weight [28] and deficits in associative memory and hyper-responsivity that model aspects of alcohol-related neurodevelopmental disorders [29]. We report here that the effects of alcohol on maternal and fetal livers are metabolically different. We demonstrate that alcohol causes an imbalance in maternal glucose and amino acid metabolism that suggests an impaired fetal utilization of these nutrients, and these correlate with, and may contribute to, impairments in fetal weight and organ development.

## 2. Materials and Methods

### 2.1. Animals and Diets

Five-week-old C57BL/6J female mice (Jackson Laboratories (Bar Harbor, ME, USA)), consumed the fixed-nutrient, purified diet AIN-93G (TD.94045, Envigo Teklad, Madison, WI, USA; [30]) throughout the study. At 8 weeks of age, *n* = 9 mice per treatment group were mated overnight with C57BL/6J males (Jackson Laboratories). The morning of vaginal plug detection was defined as embryonic day (E) 0.5 (E0.5). Starting at E8.5, pregnant females received 3 g/kg alcohol (ALC; 200 proof alcohol, USP grade; Decon Labs, King of Prussia, PA, USA) or isocaloric maltodextrin (CON; LoDex-10; #160175, Envigo Teklad, Madison, WI, USA), daily via oral gavage. On E17.5, two hours after the last gavage, mice were euthanized by isoflurane overdose and their tissues were flash-frozen for analysis. The maternal liver samples are 9 biological replicates per group, and fetal liver samples are the corresponding pool of four fetuses from these 9 dams. Samples were weighed at collection. All protocols were approved by the Institutional Animal Care and Use Committee of the David H. Murdoch Research Institute.

### 2.2. Data Analysis and Statistical Analysis

Untargeted metabolite analysis was performed by Metabolon (Durham, NC, USA). The detailed methods used by Metabolon for metabolite separation, detection, and alignment are presented in [28,31] and appended as supplementary methods. Briefly, the protein samples were extracted in methanol and separated using reverse phase/ultra-high-performance liquid chromatography-mass spectrometry (UPLC-MS)/MS with positive and negative ion mode electrospray ionization. Raw-data were extracted and the peaks identified based on the criteria of retention time/index, match to a mass to charge ratio +/− 10 ppm, and chromatographic data (MS/MS spectrum). Peaks were quantified using area-under-the-curve.

For data analysis, a non-parametric Mann–Whitney U-test (Wilcoxon Rank sum test) was used to identify fold-change differences between the two treatment groups. This was performed after testing for unequal variance (Shapiro–Wilks test) and normality (Levene’s test). Missing values were imputed using the minimum value obtained for that metabolite in that tissue. Any metabolite that was below the detection limit in five or more of the nine samples in each treatment group was removed from the dataset. Fold-change was determined as the abundance of metabolites in the alcohol-exposed group compared to control group (ALC/CON). *p*-Values were adjusted for multiple testing correction using the Benjamini-Hochburg False Discovery Rate (FDR) and are denoted as q-values. A critical q-value of 0.05 was used to define significant difference between the ALC and CON groups in both maternal and fetal livers and was performed in the R program (Version 3.6.3) (http://cran.r-project.org/; accessed on: 30 October 2021). Trends are noted where 0.05 < q ≤ 0.1.

To identify features that separate ALC and CON groups, multivariate analyses (principal component analysis (PCA) and partial least square discriminant analysis (PLS-DA)), data visualization (heatmap and correlation matrix) and pathway analysis were performed in MetaboAnalyst 5.0 (Online tool). Samples were entered in rows, and each metabolite (variable) was added in the columns to form a matrix, which was uploaded on the MetaboAnalyst 5.0 website (https://www.metaboanalyst.ca/MetaboAnalyst/home.xhtml; accessed on 18 December 2021). K-means clustering was performed using the factoextra package (version 1.0.7) and visualized using ggplot2 (version 3.3.5) in the R program (Version 3.6.3). PLS-DA variable importance projection (VIP) scores were used to identify each variable’s contribution to the model, using the leave-one-out cross-validation (LOOCV) method. To identify top pathways in each cluster, MetaboAnalyst 5.0 pathway analysis tool was used with Kyoto Encyclopedia of Genes and Genomes (KEGG) IDs as input, selecting the following parameters for analysis: *Mus musculus* library (version Oct2019) as pathway library, “Globaltest” for pathway enrichment, and “relative betweenness centrality” as a measure for topological analysis. To identify maternal–fetal metabolites that are predictive of fetal phenotypic outcomes, we performed ortho-PLSDA (oPLSDA) in the R program. The analysis was performed with the ropls package (version 1.20.0) using scaled z-score metabolite values as predictors and the numerical values of the fetal phenotype for the response. Random permutation of response labels was used to achieve the highest predictive accuracy for the model. The first orthogonal component was extracted as VIP scores, and the top 30 were used to identify potential pathways and metabolites affecting the phenotypic outcome. Pearson’s correlation analysis was performed using ggpubr package (version 0.4.0) in R. The boxplots and correlation analysis plots were generated using ggplot2 package in R with metabolite abundance of CON set to 1.

## 3. Results

### 3.1. Litter Characteristics

The maternal gestational weight gain, litter size, percent survival, and fetal weight outcomes were previously reported in [28] and are shown in Appendix A. No changes in maternal weight gain across pregnancy, litter size, or percent survival were identified in this study. Although fetal body weight was not different between the two groups, alcohol exposure significantly reduced the fetal liver weight (ALC vs. CON, 0.040 ± 0.004 g vs. 0.049 ± 0.004 g; *p* < 0.001), fetal brain weight (ALC vs. CON, 0.044 ± 0.003 g vs. 0.052 ± 0.004 g; *p* < 0.001), and fetal liver weight: body weight ratio (ALC vs. CON, 0.045 ± 0.005 vs. 0.053 ± 0.004; *p* < 0.01), and trended to decrease fetal brain weight: body weight ratio (ALC vs. CON, 0.050 ± 0.007 vs. 0.056 ± 0.004; *p* = 0.07).

### 3.2. Metabolite Profiles Distinctly Separate ALC and CON Dams but Not Their Fetuses

Untargeted metabolomics analysis identified a total of 854 unique metabolite biofeatures in the livers of alcohol-exposed dams and fetuses and their controls. Of these, 66 metabolites were uniquely detected in only maternal (*N* = 44) or fetal (*N* = 22) liver and an additional 64 were detected in fewer than five of nine maternal or fetal samples; these were excluded from further analysis. This left a total of 724 metabolites held in common between maternal and fetal liver for analysis. In the dams, 191 metabolites (26.4%) had significantly altered abundance in ALC versus CON; of these, 115 (60.2%) had increased abundance and 76 (39.8%) were decreased (q ≤ 0.05; Appendix A; Appendix A). In contrast, only 31 metabolites (27 increased; 4 decreased) were significantly altered in ALC fetuses compared to CON (Appendix A; Appendix A). Consideration of those metabolites having a q-value greater than 0.05 and less than 0.10 (0.05 < q-value ≤ 0.10) identified an additional 47 metabolites (6.5%; 26 increased, 21 decreased) in maternal liver and 42 metabolites (5.8%; 26 increased, 16 decreased) in ALC fetuses compared to their respective controls. This suggested that alcohol exposure had a significantly greater impact upon hepatic activity within maternal liver as compared with the fetus, and was consistent with liver being a primary target of orally administered alcohol through its entry via the hepatic portal circulation [22]. In contrast, fetal hepatic activity appeared to be minimally impacted at this level of alcohol exposure (3 g/kg).

Multivariate analysis of the 724 common metabolites clearly separated the hepatic metabolite profiles of ALC dams from CON. In the PCA (Figure 1a), PC1 explained 22.7% of the variance between ALC and CON dams and PC2 12.9% of the variance. The scree plot and 3D-plot of metabolites is shown in Appendix A. We did not obtain additional separation of metabolites in PC3 of the 3D-plot, and thus we did not explore additional PCs for analysis, as they did not further inform the data. The PLSDA (Figure 1b) (R^2^ = 0.9437; Q^2^ = 0.8503) and K-means clustering (Figure 1c) both identified a similar separation and variance between the ALC and CON metabolite profiles. Heatmap clustering (Figure 1d) and Spearman correlation of samples (Figure 1e) affirmed the strong correlation within the CON and within the ALC hepatic metabolites, and not with the between-group comparison. In contrast, the fetal liver metabolite profiles could not be distinguished using the unsupervised multivariate methods, PCA (Figure 1f; Appendix A) and K-means clustering (Figure 1h), and in this instance treatment did not account for the variance on PC1 but may have contributed to other components. We explored additional components (PC3 in the 3D-plot; Appendix A) but the separation along PC3 is diagonal, instead of vertical, as one would expect for clear visual separation of two groups (such as ALC vs. CON). Other factors, such as timing of tissue collection post gavage, fetal sex, and group housing, did not contribute to the variance in PC1 either. The supervised PLSDA (Figure 1g), on the other hand, separated the CON and ALC fetal hepatic profiles but treatment accounted for only 15% of the variance in PC1 (R^2^ = 0.8797; Q^2^ = 0.5911). This latter may reflect biases inherent to the PLSDA that already takes treatment groups into account. Heatmap clustering (Figure 1i) and Spearman correlation of samples (Figure 1j) also affirmed the similarity of the ALC and CON fetal hepatic profiles. Overall, these data suggest that the maternal and fetal livers respond quite differently to the alcohol exposure, with clear differences that distinguish ALC and CON maternal liver, and lesser effects of ALC upon fetal liver.

### 3.3. Amino Acid Catabolites Are Enriched in Alcohol-Exposed Maternal and Fetal Liver

As a part of the unbiased, untargeted analysis and to inform the alcohol-mediated differences in the maternal and fetal hepatic profiles, we analyzed the 724 metabolites in the PLSDA model using variable importance projection (VIP) score plots. Figure 2 shows the top 30 metabolites that contributed to the variance in the PLSDA. For maternal liver (Figure 2a), the most common metabolites enriched in ALC dams were lipid-related (12 of 30) and included diacylglycerols, phospholipids, and free fatty acids, most (83.3%) of which were increased. Amino acid catabolites comprised the next most abundant group (16.67%) and included urea, alpha-hydroxyisovalerate, pipecolate, and picolinate, all of which were increased in ALC. Metabolites in the pentose phosphate (6-phosphogluconate, arabitol) and glycolytic (fructose-6-phosphate) pathways were also enriched in the ALC maternal liver.

For fetal liver, parallel analysis was performed with these 724 metabolites. The top 30 metabolites from the VIP scores were distinct from those identified in maternal liver apart from four metabolites: urea, arabitol/xylitol, sulfate, and alpha-hydroxyisovalerate (Figure 2b). Many of the fetal metabolites were amino acid-related (36.67%) and included N-acetylated amino acids (N-acetylglycine, N-acetylthreonine, and N-acetyltaurine) and catabolites of essential amino acids (alpha-hydroxyisovalerate, saccharopine, 1-methylhistamine, and 3-hydroxyisobutyrate). Additional metabolites were related to lipids (16.67%) and cofactors/vitamins, sugars, nucleotides, and unknowns (each ≤6.7%). Thus, although the hepatic metabolite profiles of ALC dams and their fetuses largely differed, amino acid catabolites contributed to the separation of the ALC metabolite profiles within PC1 of the PLSDA in both.

### 3.4. Maternal Metabolite Profile Is Consistent with the Known Impact of Alcohol on Hepatic Metabolism

To gain mechanistic insight into alcohol’s impact upon maternal and fetal metabolism, we mapped the metabolites having significantly altered abundance using a Metaboanalyst. Only 85 of 191 significantly altered maternal metabolites were annotated in KEGG, and due to this low annotation the number of hits for each pathway was low. Regardless, the majority of the pathways that emerged were consistent with alcohol’s known effects on hepatic metabolism and included pentose phosphate and nicotinamide pathways, glycolysis, and gluconeogenesis, and fatty acid synthesis (Table 1). For fetal liver, the number of significantly altered metabolites was low and only half were annotated in KEGG (14 of 31); thus, their pathway analysis was not feasible.

To overcome these limitations in annotation, we visually inspected the metabolites in major biochemical pathways. With respect to glycolysis and the TCA cycle, in the dams (Figure 3a) alcohol exposure significantly reduced the hepatic abundance of glucose (0.79-fold), glucose-6-phosphate (0.66-fold), and fructose-6-phosphate (0.51-fold). However, these metabolite changes were not mirrored in the livers of their fetuses, where their abundance did not differ except for fructose-6-phosphate, which trended to decrease in ALC fetal liver compared to CON fetus (q = 0.09; Appendix A).

With respect to lipids, alcohol’s impact on maternal hepatic metabolism was consistent with its known effects in non-pregnant adults [21]. ALC dams had significantly increased the abundance of medium chain fatty acids (FA) (1.97-fold; q = 4.1 × 10^−5^), long chain saturated FA (1.14-fold; q = 0.04), and long chain monounsaturated FA (1.45-fold; q = 0.001) (Figure 3b). Several fatty acyl carnitines were reduced by alcohol, and these were mostly short-chain lipids (C2–C7) (Table 2). With respect to complex lipids, only phosphatidylglycerol-related species were significantly increased compared to controls. In contrast, the lipid profile in fetal liver remained unchanged (Appendix A).

We next evaluated the hepatic abundance of amino acids and their related metabolites. ALC dams had significant reductions in select essential amino acids, including isoleucine (0.88-fold), methionine (0.89-fold), phenylalanine (0.92-fold), and tryptophan (0.86-fold), the non-essential amino acid asparagine (0.89-fold), and the essential-derived amino acids tyrosine (0.87-fold) and cysteine (0.59-fold; Figure 4); in contrast, histidine was significantly elevated (1.39-fold). Importantly, hepatic urea levels were significantly increased (2.07-fold) in ALC dams along with citrulline (1.42-fold), perhaps indicating increased activity of the urea cycle. This was accompanied by enrichments of several N-acetylated amino acids and catabolites of essential amino acids including their carnitine derivatives (Table 2). Alcohol exposure had little impact on the hepatic amino acid abundance in the fetuses, apart from a significant rise in their hepatic urea (1.95-fold; q = 4.94 × 10^−3^; Appendix A).

With respect to purine and pyrimidine metabolism, maternal ALC liver had significant elevations in thymine (1.38-fold; q = 0.01), guanosine (1.22-fold; q = 0.03), cytidine (1.24-fold; q = 0.02), and CMP (4.94-fold; q = 0.01), and significant reductions in the purine catabolites xanthine (0.90-fold; q = 4.33 × 10^−3^), urate (0.50-fold; q = 0.01), and allantoin (0.68-fold; q = 1.41 × 10^−3^; Appendix A), and these were unaffected by alcohol in fetal liver (Appendix A).

### 3.5. Maternal Hepatic Metabolites Are Predictive of Fetal Phenotypic Outcomes

To gain insights into how these maternal hepatic changes might inform the phenotypic outcomes of their fetuses, we used oPLSDA to identify the metabolite drivers in maternal liver (*N* = 724 metabolites) that defined the separation of treatment groups with respect to fetal body weight, liver weight, and brain weight.

With respect to fetal body weight, PCA separated the maternal liver metabolites overlayed with treatment groups with a variance of 23% in Component 1 and 13% in Component 2 (Appendix A). However, the separation of metabolites with respect to fetal body weights was not distinct using PLSDA (Component 1 = 8%; Component 2 = 18%) (R^2^ = 0.442; Q^2^ = −1010), or oPLSDA (t1 = 6%) (R^2^ = 0.255; Q^2^ = −15.2) models. Identification of the top 30 maternal metabolites in Component 1 of oPLSDA revealed that the majority of the metabolites (20 of 30) were lipid-related compounds (mainly stearoyl-related), and these were largely not affected by alcohol treatment (Appendix A). Only 4 of the 30 metabolites were significantly altered by alcohol at q ≤ 0.1: creatine, malonate, 1-methylnicotinamide, and 1-linoleoyl-GPE (18:2). This lack of treatment effect likely reflects that the mean fetal body weights for the ALC and CON groups did not significantly differ (Appendix A), and both had a similar range of high and low values, and suggests that these metabolites are predictors of fetal body weight regardless of treatment. Pearson’s correlation analysis of the top six metabolites revealed that hepatic 3′AMP (R = −0.55, *p* = 0.018) and the bile acid β–muricholate (R = −0.52, *p* = 0.026) and its tauro-derivative (R = −0.45, *p* = 0.058) were negatively correlated with fetal weight, whereas the lipids N-stearoyl-sphingosine (d18:1/18:0) (R = 0.51, *p* = 0.031), stearoyl-linoleoyl-glycerol (18:0/18:2) (R = 0.62, *p* = 0.006), and 1-stearoyl-2-linoleoyl-GPE (R = 0.62, *p* = 0.006) were positively correlated with fetal body weight (Appendix A), suggesting they also may be predictors of fetal weight irrespective of treatment.

The separation of maternal metabolites based on fetal liver weights (Figure 5a), revealed distinct separation of ALC and CON treatment groups using PCA (Component 1 = 23%; Component 2 = 13%), PLSDA (Component 1 = 22%; Component 2 = 9%) (R^2^ = 0.444; Q^2^ = −1230) and oPLSDA (t1 = 14%) (R^2^ = 0.302; Q^2^ = −13.7) analyses. The top 30 maternal liver metabolites identified by oPLSDA that separate fetal liver weights as per treatment are shown in Figure 5b. Of these, 36.67% were lipid-related (11 of 30), 23% were carbohydrates (7 of 30), and only 10% were amino-acid-related (3 of 30). Of these 30, 27 metabolites were significantly altered by alcohol treatment at q ≤ 0.1. Lipid-related metabolites included several phospholipids, although some were increased and others decreased. Carbohydrate-related metabolites included enrichments in neuraminic sugars (N-acetylneuraminate and cytidine 5’-monophospho-N-acetylneuraminic acid) and significantly reduced abundance of the pentose-phosphate pathway intermediates arabitol, sedoheptulose-7-phosphate, and 6-phosphogluconate. Both neuraminate metabolites, as well as nicotinate ribonucleoside and dioleoyl-GPC, negatively correlated with fetal liver weight (R = −0.59 to −0.75; *p* < 0.001; Figure 5b,c) and suggests their potential as predictors of maternal alcohol exposure and reduced fetal liver weights.

Similar to fetal liver outcomes, the PCA (Component 1 = 23%; Component 2 = 13%), PLSDA (Component 1 = 22%; Component 2 = 7%) (R^2^ = 0.433; Q^2^ = −1230) and oPLSDA (t1 = 14%) (R^2^ = 0.285; Q^2^ = −14.6) analyses distinctly separated the maternal metabolites overlayed with treatment groups based on fetal brain weights (Figure 6a). All but two of the top 30 metabolites that separated the fetal brain weights in oPLSDA were significantly altered by alcohol-exposure (Figure 6b), and they included metabolites related to amino-acids and lipids (26%), carbohydrates (20%), and nucleotides (13%). Several of these strongly correlated with fetal brain weight, with S-adenosylhomocysteine (SAH), urea, and kynurenate having negative correlation (R = −0.63 to −0.70; *p* < 0.005), and glucose-6-phosphate and methionine being positively correlated (R = 0.71 and 0.76, respectively, *p* = 0.001) (Figure 6c). For all associations of maternal metabolites with fetal body weight, liver weight, or brain weight, the top 30 metabolites were unique to each phenotypic outcome except for citrulline, 3-hydroxyoleate, and 6-phosphogluconate, which were common drivers for both fetal liver weight and brain weight but not fetal body weight.

The low R^2^ and Q^2^ values provided for the PLSDA and oPLSDA models suggest they weakly model predictive biosignatures. Instead, we applied the oPLSDA results to correlation analysis (For result Section 3.5 and Section 3.6) to explore candidate associations between the top metabolites and fetal outcomes, and thus gain insight for future mechanistic studies.

### 3.6. Fetal Hepatic Metabolites Predict Fetal Brain Weight but Not Fetal Body and Liver Weight

We then investigated whether fetal hepatic metabolites (*N* = 724 metabolites) similarly predicted fetal phenotypic outcomes, and whether they could separate the treatment groups based on the fetal body, liver, and brain weight phenotypes.

The separation of fetal liver metabolites based on fetal body weights could not be achieved using the PCA (Component 1 = 20%; Component 2 = 16%), PLSDA (Component 1 = 18%; Component 2 = 13%) (R^2^ = 0.479; Q^2^ = −320) and oPLSDA (t1 = 15%) (R^2^ = 0.311; Q^2^ = −15) analyses (Appendix A). The oPLSDA analysis identified the top 30 metabolites in Component 1 predictive of fetal body weights but none were affected by alcohol exposure (Appendix A). Similarly, fetal liver metabolites did not separate based on fetal liver weights in the PCA and PLSDA (R^2^ = 0.521; Q^2^ = −337), though some separation could be achieved by oPLSDA (R^2^ = 0.269; Q^2^ = −14.6), as 19 of the top 30 were alcohol-responsive (Figure 7a). The majority of these top 30 metabolites (Figure 7b) were amino-acid-related (17 of 30) and included N-acetylated amino acids and essential amino acid catabolites, the rest were lipid- and carbohydrate-related. Only arabitol and alpha-hydroxyisovalerate were commonly held between the maternal and fetal metabolite sets that predicted fetal liver weight. For the top three amino-acid-related compounds, all three were negatively correlated with fetal liver weight (R = −0.53 to 0.82; Figure 7b,c) in the Pearson’s correlation analysis.

Based on fetal brain weights, the separation of fetal hepatic metabolites was achieved in Component 1 of the PLSDA (R^2^ = 0.496; Q^2^ = −407) and oPLSDA (R^2^ = 0.229; Q^2^ = −14) models, but not in the PCA (Figure 8a). Eighteen of the top 30 metabolites identified by oPLSDA were alcohol-responsive and 13 were shared with the fetal liver weight dataset. Twenty were amino-acid-related metabolites and eight of those were not affected by alcohol exposure (Figure 8b). For the top four metabolites, in the Pearson’s correlation analysis imidazole propionate was positively correlated with fetal brain weight (R = 0.61, *p* = 0.01), whereas urea (R = −0.46, *p* = 0.005), S-adenosylmethionine (R = −0.48, *p* = 0.044), and its catabolite 5-methyladenosine (R = −0.47, *p* = 0.047) were negatively correlated (Figure 8c).

## 4. Discussion

The most important finding from this study is that gestational alcohol exposure significantly alters the hepatic profile of maternal metabolites related to macronutrient metabolism, with a notable decline in glucose and elevation of amino-acid catabolites. Moreover, these changes were strongly associated with reduced fetal somatic and organ growth, suggesting these alterations significantly disrupted nutrient availability for fetal development. To our knowledge, this is the first study to investigate hepatic activity through the metabolite profiles of the maternal–fetal dyad in a model of FASD. As liver is the major regulator of nutrient availability for both mother and fetus [16], this study provides mechanistic insights into the maladaptations caused by alcohol consumption in pregnancy and their potential consequences to maternal and fetal metabolism and fetal development.

It is perhaps not surprising that alcohol has quite divergent effects on maternal and fetal hepatic metabolism, with the greater impact on maternal hepatic metabolism. An obvious explanation for the differential response is that maternal and fetal hepatic activities are quite distinct. The adult liver is fully developed to perform its metabolic functions, and is the primary site for gluconeogenesis, glycolysis, and glycogen metabolism to maintain blood glucose levels. Lipid synthesis and secretion is also a critical function, as liver utilizes fatty acids for energy purposes. Finally, adult liver contributes to protein synthesis and breakdown, as well as disposal of nitrogenous waste via the urea cycle [17]. In contrast, fetal liver is metabolically inefficient and under-developed. In the fetal mouse between E8.5 to E17.5 (the alcohol exposure window), the liver grows rapidly and is a primary site for hematopoiesis. At E17.5, the fetal liver cells start differentiating and expressing genes for fatty acid and glucose metabolism [32,33]; however, gluconeogenesis is not activated in the fetus until after birth [34]. Only lipogenesis and glycogen metabolism are active in the fetus, to help maintain growth and energy reserves for after-birth utilization [34,35]. Thus, fetal metabolic processes are quite distinct from those of its mother, and it must rely on her for those missing processes. We previously reported that, in a normal pregnancy, maternal and fetal hepatic profiles differ significantly [31]. The fetus has reduced hepatic glucose and essential amino acid pools compared to its mother, and it depends entirely on maternal pools for the same. Fetal hepatic pools are enriched in serine, glycine, aspartate, and glutamate, which reflects their due contributions to endogenous nucleotide synthesis and protein synthesis [31]. Finally, with respect to alcohol metabolism, although the mother and fetus experience similar alcohol concentrations [36,37,38] the fetal liver has lower abundance of P450 enzymes (approx. half of that of adult liver) and alcohol dehydrogenase activity (4 to 5% of adult activity in ewes and rats and 10% of adult levels in the human fetus) [37,38,39,40]. Thus, the fetus largely depends on its mother to metabolize most of its alcohol. Because of these inherent differences, along with the known impact of alcohol on hepatic metabolism, we observe substantial differences in the metabolite profiles of ALC dams compared to CON, and fewer differences in their fetuses.

The maternal–fetal hepatic metabolite profiles described here inform alcohol’s impact on maternal–fetal nutrient needs. To our knowledge, this is the first study to document a significant reduction of hepatic glucose content and select glycolytic intermediates in response to gestational alcohol exposure, changes that strongly suggest an alteration in the glycolytic flux in ALC dams. Because the main source of fetal glucose is maternal plasma pools, decreased maternal hepatic glucose may indicate a reduced capacity of the dam to supply glucose to her fetus. This decline is accompanied by reductions in amino acid content and enhanced abundance of amino acid catabolites, N-acetylated amino acids, and urea levels in ALC dams, changes that may reflect a compensatory attempt to meet this glucose gap through increased protein turnover and release of amino acids for gluconeogenesis. The elevations in glycerol and free fatty acids may, in part, be an additional attempt to address this gap, supplying glycerol for gluconeogenesis and fatty acids for maternal energy needs. We further note that fetal hepatic glucose and amino acid levels are unchanged in this model. There are two non-mutually exclusive possible explanations: 1. The fetus is able to draw sufficient glucose and amino acids for its anabolic needs from maternal resources, although this likely imposes stress on maternal metabolism. 2. The fetus utilizes its own resources that would otherwise be allocated for growth, hence the lower organ weights in this model (Appendix A). Previous studies from our lab have demonstrated that reducing the maternal protein intake in alcohol exposed dams hugely impact fetal development in a sex-specific manner [41]. In the non-pregnant state, alcohol consumption is associated with reduced hepatic gluconeogenesis [42,43,44] and increased fatty acid and triglyceride synthesis [45]. Our gestational data are largely consistent with this literature and further suggest that alcohol imposes a catabolic load that forces the dam to utilize alternative fuels to meet the exceptional energy demands of her highly anabolic pregnancy state.

Correlation analyses suggest that the alterations in maternal hepatic metabolites have negative consequences for fetal development. We identify maternal and fetal metabolites that separately correlate with fetal body weight, liver weight, and brain weight. That these metabolite sets are distinct for fetal brain versus liver versus body weight is consistent with demonstrations that, under nutritional stress, the fetus can prioritize growth of critical organs, such as brain at the expense of body and visceral organ growth, by altering the allocation of nutrients [46,47]. Interestingly, it was the glycolytic/gluconeogenic intermediates, glucose-6-phosphate and fructose-6-phosphate, rather than glucose itself that significantly correlated with fetal brain weight, highlighting this pathway as a strong influence on its growth. That these intermediates may represent gluconeogenesis is supported by the additional inverse correlation of both fetal liver and brain weight with urea and essential amino acid catabolites, including alpha-hydroxyisovalerate, kynurenate, and picolinate. The elevations in numerous N-acetylated amino acids point to protein turnover as an important source for these catabolized amino acids. That these maternal metabolites are such strong predictors for fetal organ weights serves to highlight the critical role played by maternal hepatic activity in supporting fetal development, a contribution that has heretofore been largely under-investigated in considering alcohol’s adverse impact upon the fetus.

In addition, the metabolites that correlated with fetal liver and brain weights were pentose phosphate pathway (PPP) intermediates, including 6-phosphogluconate, sedoheptulose-7-phosphate, and arabitol, all of which were reduced by alcohol in maternal liver and were positively correlated with fetal weight. The PPP generates nicotinamide adenine dinucleotide phosphate (NADPH) reducing equivalents as an alternative route for glucose oxidation, and produces ribose-5-phosphate for nucleotide synthesis and sugar phosphates for amino acid synthesis, both of which can also shunt carbon back to the glycolytic pathway [48]. The NADPH produced by this pathway also serves as a reducing agent for anabolic pathways, such as fatty acid synthesis, and to regenerate reduced glutathione in response to oxidative stress [49,50]. Alcohol’s impact on the PPP is not well characterized, with suggestions that alcohol inhibits the PPP in non-pregnant alcoholics and in alcoholic liver disease [51,52,53]. The brain content of PPP intermediates is reduced in rats exposed to excessive alcohol drinking [54], suggesting that effects are conserved across organs. A reduction of PPP activity and its generated NADPH would be predicted to enhance oxidative stress and, consistent with this, reduced glutathione (GSH) was decreased in maternal liver (0.58, q = 0.049) whereas oxidized glutathione (GSSG) was unaffected (0.95, q = 0.101) thereby decreasing the GSH/GSSG ratio. In light of these observations, the potential impact of these PPP intermediates upon both maternal metabolism and fetal development requires further investigation.

Another relevant metabolic process to emerge from this analysis is one-carbon metabolism. Supplementation with the methyl donor choline confers growth and behavioral benefits to the alcohol-exposed offspring [3,4,55], although the underlying mechanism(s) remain unclear. Alcohol inhibits hepatic enzymatic activities in the methionine cycle [56], changes that contribute to progressive liver damage, including increased fat deposition, endoplasmic reticulum-mediated stress, accumulation of damaged proteins, and apoptosis [57], and such disruptions would impair the liver’s ability to generate nutrients for the developing fetus. Pathways involving choline, methionine, and folate metabolism are also essential to healthy fetal brain development, and deficits during pregnancy are associated with impaired memory and learning in the offspring [58]. We and others recently reported that choline supplementation improves fetal body, liver, and brain weights in rodent models of FASD [59,60], and consistent with this, we find here that the maternal hepatic content of the one-carbon metabolites methionine and S-adenosylhomocysteine (SAH) are positively correlated with fetal brain weight. Similarly, although alcohol affected the hepatic content of far fewer metabolites in the fetus, the methyl donor S-adenosylmethionine (SAM) and its catabolite 5-methylthioadenosine are negatively correlated with fetal brain weight. Although counterintuitive, the hepatic content of both was elevated by alcohol, and their accumulation could signal impairment in the various methyl transferases that use SAM as a substrate. Interestingly, elevated 5-methylthioadenosine also emerged in our prior analysis of metabolites produced by alcohol-exposed human neural stem cells [61]. The association of these metabolites with fetal brain weight affirms the critical importance of one-carbon metabolism to fetal development, and offers insight into how alcohol might affect choline-dependent activities in the maternal–fetal dyad.

Finally, serum sialic acid (N-or O-derivatives of the sugar neuraminic acid) is an established marker of alcohol consumption and liver disease [62,63,64], and in these dams N-acetylneuraminate showed one of greatest fold-changes in response to alcohol (FC 26.67, q = 0.0014). Both it and cytidine 5′-monophospho-N-acetylneuraminic acid (FC 2.51, q = 0.060) were negatively correlated with fetal liver weight. Sialic acids are acetylated derivatives of neuraminic acid and contribute to glycoproteins that mediate cell-to-cell interactions and membrane receptor functions [63]. Chronic alcohol consumption impairs hepatic protein glycosylation and could potentially elevate these metabolites [64]. Importantly, elevated sialic acid is an inflammatory marker and in pregnant women is associated with gestational diabetes and type 2 diabetes mellitus [65,66]. The elevated N-acetylneuraminate is consistent with sialic acid’s role as a marker for alcohol exposure, and its negative correlation with fetal liver weight suggests it serves as a marker for fetal impairment in alcohol exposure.

In conclusion, this is the first study to investigate maternal–fetal hepatic metabolism under PAE. Although this study provides a single snapshot of liver metabolites, these findings provide mechanistic insights into metabolic disruptions through which alcohol might suppress fetal growth and development. Alcohol significantly reduced the hepatic glucose content, abundance of glycolytic intermediates, and amino acid content in the dams. These changes were accompanied by elevations in amino acid catabolites and free fatty acids, which may reflect a compensatory attempt by the dam to enhance fetal glucose availability at the expense of her own metabolic needs. These metabolic alterations correlated with fetal growth outcomes, and their association highlights the underappreciated influence of maternal hepatic metabolism on fetal outcomes. Future studies will provide mechanistic insight into these alcohol-driven metabolic disruptions and their impact on fetal development.

## Figures and Tables

**Figure 1 nutrients-14-01096-f001:**
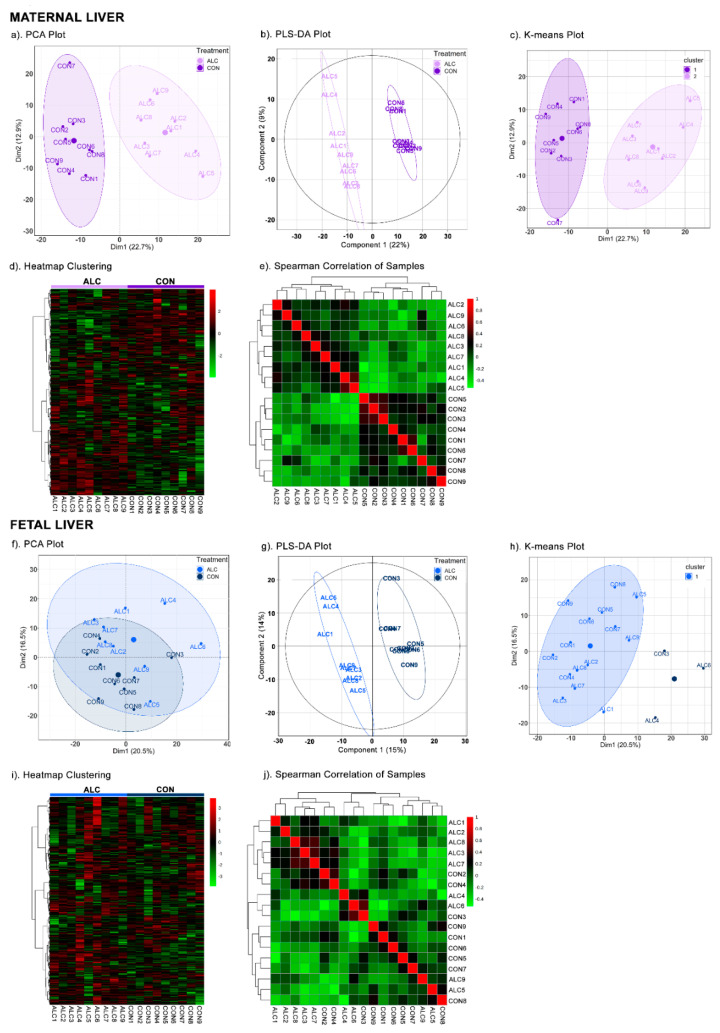
Multivariate analyses of maternal and fetal metabolite profiles. In maternal liver, PCA (**a**), PLSDA (**b**), K-means clustering (**c**), heatmap clustering (**d**), and Spearman correlation of samples (**e**) clearly separated the ALC vs. CON groups. Whereas, in fetal liver, only PLSDA (**g**) separated the ALC vs. CON groups with a variance of 15% in Component 1. PCA (**f**), K-means clustering (**h**), heatmap clustering (**i**), and Spearman correlation of samples (**j**) were unable to separate the ALC vs. CON groups. The high, unchanged to low abundance values of metabolites in both groups are represented by a gradient of red to green color in heatmaps. In correlation of samples, positive, unchanged and negative correlations are represented from the red to green color. *N* = 9 each for the CON and ALC maternal and fetal samples, *N* = 724 for the metabolites.

**Figure 2 nutrients-14-01096-f002:**
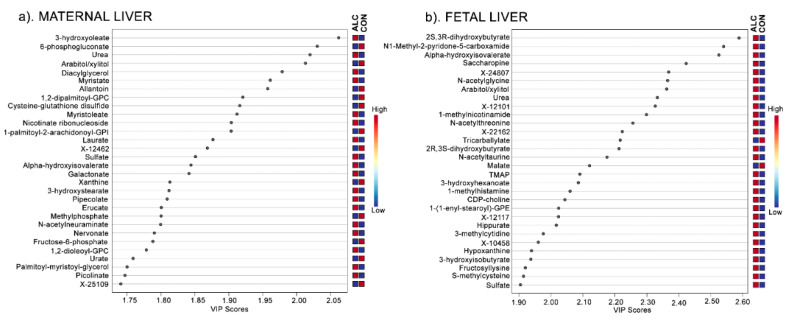
VIP Scores Plot of the top 30 metabolites in maternal liver (**a**) and fetal liver (**b**) that contribute to the separation of ALC vs. CON metabolite profiles in the Component 1 in the PLSDA. The blue and red boxes on the right indicate whether the mean metabolite abundance is increased (red) or decreased (blue) in ALC vs. CON.

**Figure 3 nutrients-14-01096-f003:**
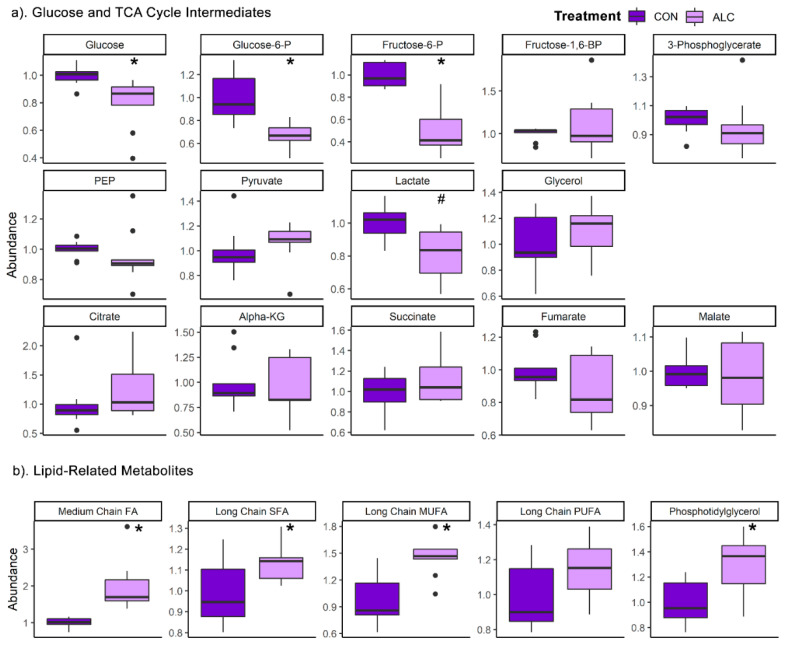
Relative abundance of (**a**) glucose and TCA cycle intermediates and (**b**) lipid classes in ALC vs. CON in maternal liver. Lipid classes are: medium chain FAs (C6:0–C12:0), saturated FAs (C14:0–C22:0), monounsaturated FAs (C14:1–C22:1), polyunsaturated FAs (C14:2–C24:6), and phosphotidylglycerol (C16:0, C18:0, C18:1, and C18:2 at sn1 or sn2 position). Abundance in CON is normalized to 1.0, and comparisons used Wilcoxon test. Boxplots depict the data’s spread (measured in inter quartile range), middle line depicts the median, and the dots indicate outliers. * q ≤ 0.05, # 0.05 < q ≤ 0.10. Alpha-KG—alpha-ketoglutarate; FA—Fatty Acid; Fructose-6-P—fructose-6-phosphate; Fructose-1,6-BP—fructose-1,6-bisphosphate; Glucose-6-P—glucose-6-phosphate; MUFA—monounsaturated FA; PEP—phosphoenolpyruvate; PUFA—polyunsaturated FA; SFA—saturated FA.

**Figure 4 nutrients-14-01096-f004:**
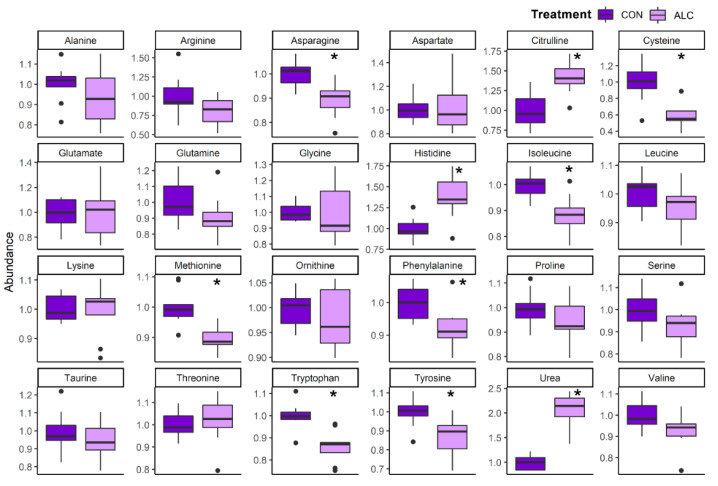
Relative abundance of amino acid metabolites in ALC vs. CON in maternal liver. Abundance in CON is normalized to 1.0, and comparisons used Wilcoxon test. Boxplots depict the data’s spread (measured in inter quartile range), middle line depicts the median, and the dots indicate outliers. * q ≤ 0.05.

**Figure 5 nutrients-14-01096-f005:**
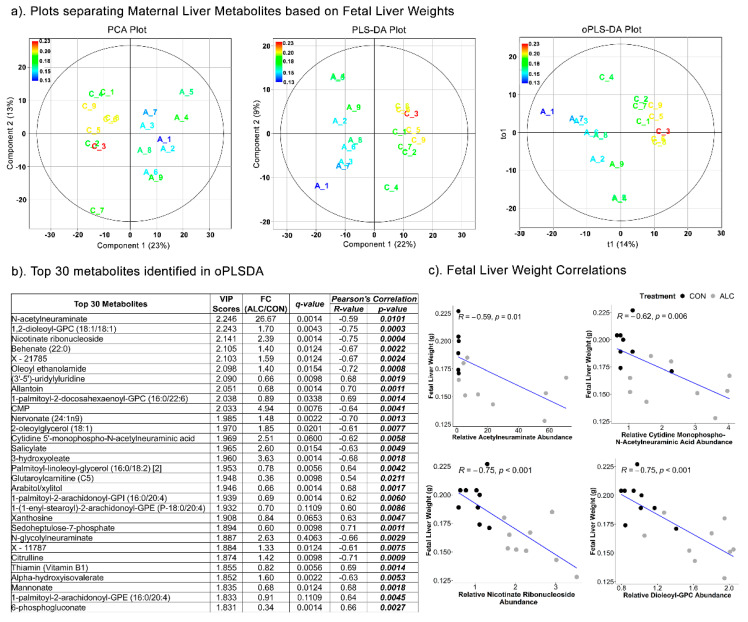
Separation of maternal liver metabolites as ALC vs. CON based on fetal liver weights. (**a**) PCA, PLS-DA, and oPLSDA plots depicting the separation of maternal liver metabolites as ALC vs. CON based on fetal liver weights. The legend and the corresponding color in the plots indicate increase in fetal liver weights (in g) values from blue to red. (**b**) The top 30 metabolites extracted as VIP scores in the first orthogonal component in oPLSDA with FC (ALC/CON), q-value, and Pearson’s correlation (R) analysis with respect to fetal liver weights. Significant correlations at *p* ≤0.05 are bold and italicized. (**c**) Correlation plots of top select metabolites identified in oPLSDA with fetal liver weights. The blue trendlines indicate significant negative correlations.

**Figure 6 nutrients-14-01096-f006:**
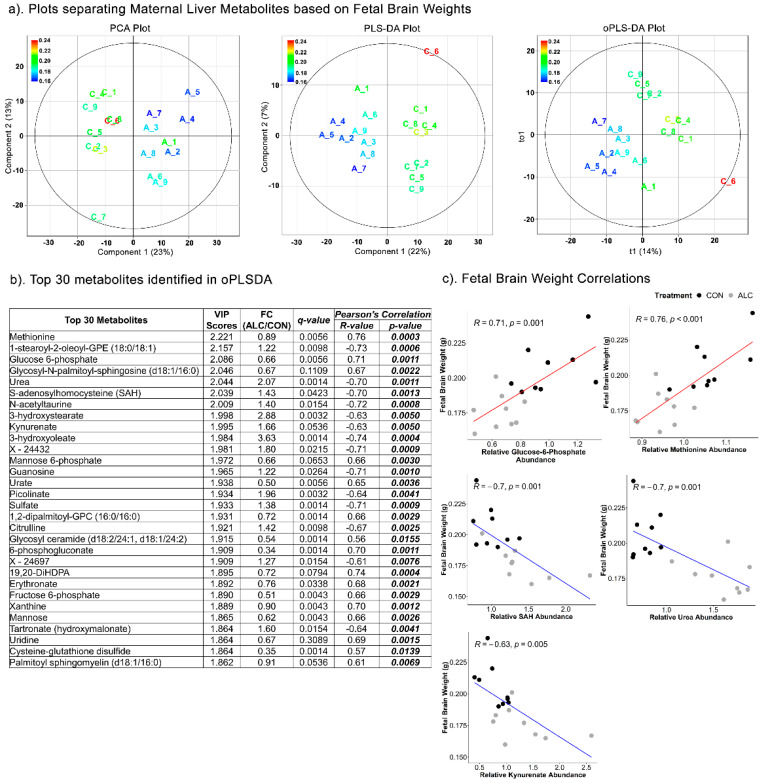
Separation of maternal liver metabolites as ALC vs. CON based on fetal brain weights. (**a**) PCA, PLS-DA, and oPLSDA plots depicting the separation of maternal liver metabolites as ALC vs. CON based on fetal brain weights. The legend and the corresponding color in the plots indicate increase in fetal brain weights (in g) values from blue to red. (**b**) The top 30 metabolites extracted as VIP scores in the first orthogonal component in oPLSDA with FC (ALC/CON), q-value, and Pearson’s correlation (R) analysis with respect to fetal brain weights. Significant correlations at *p* ≤ 0.05 are bold and italicized. (**c**) Correlation plots of top select metabolites identified in oPLSDA with fetal brain weights. The blue and red trendlines indicate significant negative and positive correlations, respectively.

**Figure 7 nutrients-14-01096-f007:**
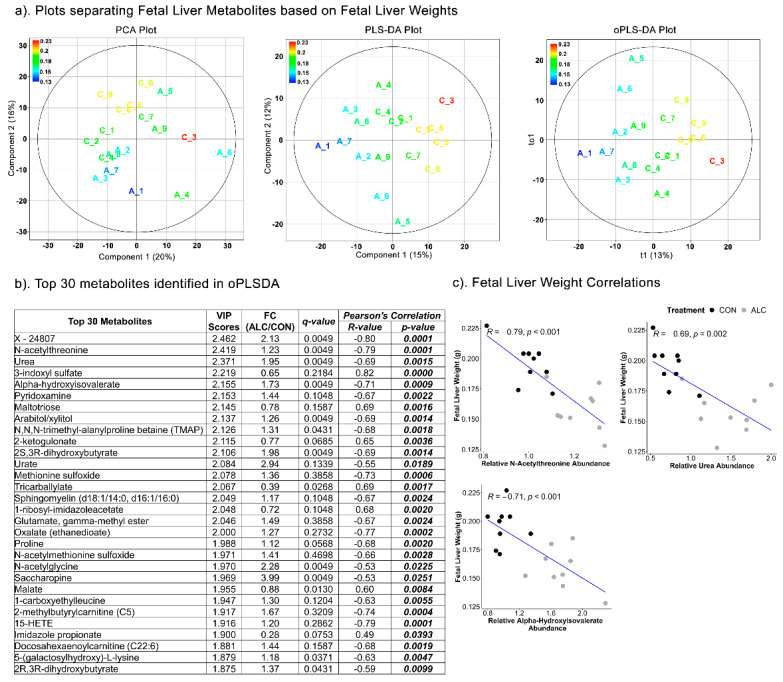
Separation of fetal liver metabolites as ALC vs. CON based on fetal liver weights. (**a**) PCA, PLS-DA, and oPLSDA plots depicting the separation of fetal liver metabolites as ALC vs. CON based on fetal liver weights. The legend and the corresponding color in the plots indicate increase in fetal liver weights (in g) values from blue to red. (**b**) The top 30 metabolites extracted as VIP scores in the first orthogonal component in oPLSDA with FC (ALC/CON), q-value, and Pearson’s correlation (R) analysis with respect to fetal liver weights. Significant correlations at *p* ≤0.05 are bold and italicized. (**c**) Correlation plots of top select metabolites identified in oPLSDA with fetal liver weights. The blue trendlines indicate significant negative correlations.

**Figure 8 nutrients-14-01096-f008:**
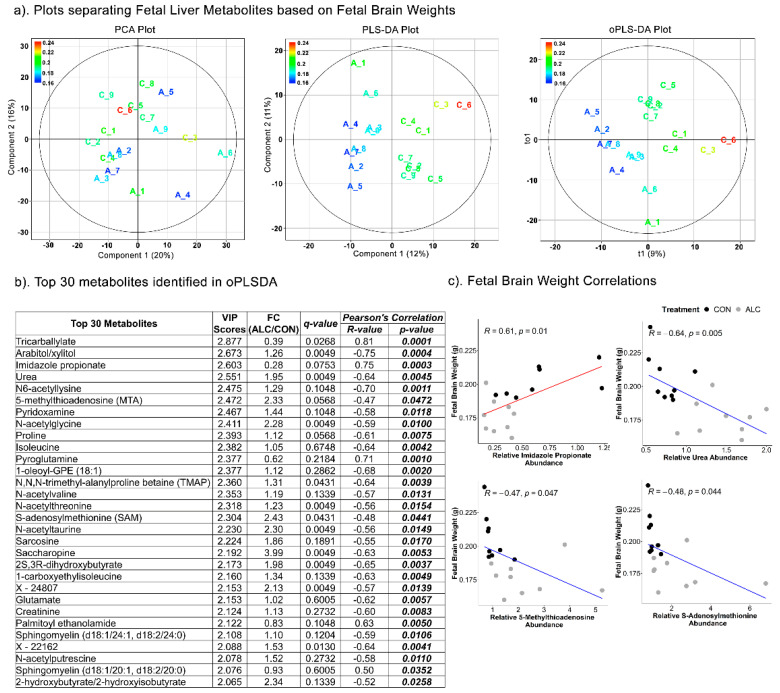
Separation of fetal liver metabolites as ALC vs. CON based on fetal brain weights. (**a**) PCA, PLS-DA, and oPLSDA plots depicting the separation of fetal liver metabolites as ALC vs. CON based on fetal brain weights. The legend and the corresponding color in the plots indicate increase in fetal brain weights (in g) values from blue to red. (**b**) The top 30 metabolites extracted as VIP scores in the first orthogonal component in oPLSDA with FC (ALC/CON), q-value, and Pearson’s correlation (R) analysis with respect to fetal brain weights. Significant correlations at *p* ≤0.05 are bold and italicized. (**c**) Correlation plots of top select metabolites identified in oPLSDA with fetal brain weights. The blue and red trendlines indicate significant negative and positive correlations, respectively.

**Table 1 nutrients-14-01096-t001:** Top pathways enriched with metabolites having significantly altered abundance in alcohol-exposed maternal liver, as identified by the pathway analysis using MetaboAnalyst.

Top Pathways	Total Compounds	Hits	Raw *p*	FDR	Metabolites Identified
Purine metabolism	66	6	4.01 × 10^−10^	1.80 × 10^−8^	Xanthine, urate, allantoin, guanosine, sulfate, urea
Pentose phosphate pathway	22	6	8.11 × 10^−10^	1.83 × 10^−8^	Glucose 6-phosphate, sedoheptulose 7-phosphate, fructose 6-phosphate, 6-phosphogluconate, gluconic acid, glycerate
Pyrimidine metabolism	39	4	2.21 × 10^−8^	3.31 × 10^−7^	3-ureidopropionate, cytidine, CMP, thymine
Nicotinate and nicotinamide metabolism	15	3	4.40 × 10^−7^	3.96 × 10^−6^	Quinolinate, nicotinate D-ribonucleoside, nicotinamide-beta-riboside
Glycolysis/gluconeogenesis	26	3	3.12 × 10^−6^	2.00 × 10^−5^	Thiamin diphosphate, fructose 6-phosphate, glucose 6-phosphate
Fatty acid biosynthesis	47	3	7.81 × 10^−6^	4.39 × 10^−5^	Tetradecanoic acid, dodecanoic acid, decanoic acid

**Table 2 nutrients-14-01096-t002:** List of amino acid catabolite products and fatty acyl carnitines in alcohol-exposed maternal liver with fold change (FC) and q-values. q ≤ 0.10 is bold.

Metabolite Name	FC (ALC/CON)	*p*-Value	q-Value
N-Acetylated Amino-acid Products			
N-Acetylalanine	0.82	0.0142	**0.0536**
N-Acetylasparagine	1.49	0.0188	**0.0653**
N-Acetylaspartate (NAA)	0.93	0.7304	0.8229
N-Acetylglutamate	1.92	0.0056	**0.0264**
N-Acetylglutamine	1.24	0.3401	0.4994
N-Acetylglycine	1.35	0.7304	0.8229
N-Acetylserine	0.93	0.2224	0.3773
N-Acetylthreonine	1.29	0.0106	**0.0423**
N-Acetylhistidine	1.49	0.0106	**0.0423**
N-Acetylleucine	1.02	0.9314	0.9512
N-Acetylvaline	0.98	0.8633	0.9140
N-Acetylcysteine	0.99	0.7962	0.8632
N-Acetylmethionine	0.85	0.0503	0.1334
N-Acetyltaurine	1.40	0.0028	**0.0154**
N-Acetylphenylalanine	1.15	0.1135	0.2443
Essential Amino-Acid Catabolites			
2-hydroxy-3-methylvalerate	1.43	0.0315	**0.0961**
3-hydroxyisobutyrate	1.39	0.0625	0.1582
4-methyl-2-oxopentanoate	0.53	0.2868	0.4505
α-hydroxyisovalerate	1.60	0.0001	**0.0022**
β-hydroxyisovalerate	0.98	0.8633	0.9140
β-hydroxyisovaleroylcarnitine	1.10	0.0400	0.1109
Isovalerylcarnitine (C5)	1.40	0.1903	0.3429
Isovalerylglycine	1.46	0.2581	0.4063
Pipecolate	1.67	0.0000	**0.0014**
Picolinate	1.96	0.0002	**0.0032**
Fatty-Acyl Carnitines			
Carnitine	0.94	0.5457	0.6749
Adipoylcarnitine (C6-DC)	0.36	0.0062	**0.0287**
3-Methyladipoylcarnitine (C7-DC)	0.66	0.0000	**0.0014**
(R)-3-hydroxybutyrylcarnitine	0.53	0.0106	**0.0423**
(S)-3-hydroxybutyrylcarnitine	0.51	0.0078	**0.0338**
3-hydroxyoleoylcarnitine	1.98	0.0079	**0.0343**
Acetylcarnitine (C2)	0.87	0.2581	0.4063
Propionylcarnitine (C3)	0.67	0.1575	0.3089
Malonylcarnitine	0.65	0.0171	**0.0637**
Isobutyrylcarnitine (C4)	0.69	0.0625	0.1582

FDR: False Discovery Rate; CMP: Cytidine Monophosphate.

## Data Availability

All the processed data is available as manuscript tables/figures or supplementary information.

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
