# Peer review of "Untargeted Metabolome Analysis Reveals Reductions in Maternal Hepatic Glucose and Amino Acid Content That Correlate with Fetal Organ Weights in a Mouse Model of Fetal Alcohol Spectrum Disorders"

_nutrients, 2022, doi:10.3390/nu14051096_

Round 1

Reviewer 1 Report

Saini et al. presented a study on the fascinating field of Fetal Alcohol Spectrum Disorders. They first performed an untargeted metabolome analysis to investigate the correlation between macromolecules and fetal symptoms (fetal body, liver, and brain weight).

C57BL/6J pregnant females received 3g/kg alcohol or isocaloric maltodextrin daily. Mice were killed, organs were weighed, and untargeted metabolite analysis was performed.

Results were full-bodied, and each metabolite found was described.

The statistical analysis led to interesting correlations.

To summarize, alcohol exposure significantly altered the hepatic profile of maternal metabolites, with a notable decline in glucose and elevation of amino acid catabolites. Moreover, these changes were strongly associated with reduced fetal somatic and organ growth.

The authors elegantly address their research question.

The study is well contextualized, and the introduction sets the stage.

The figures and tables are clear and readable. Moreover, they are captions complete and accurate for supporting the findings.

Methods are well described and reproducible, adequate for the research question. Data are interpreted accurately, so the results support the conclusions.

All protocols were approved by the Institutional Animal Care and Use Committee of the David H. Murdoch Research Institute.

This research was co-funded by the National Institute of Health/National Institute on Alcohol Abuse and Alcoholism.

Author Response

We thank Reviewer 1 for the positive and constructive comments. No concerns were raised by the reviewer.

Reviewer 2 Report

The manuscript described an untargeted metabolomics on the late-term maternal and fetal livers in a C57BL/6J mouse model of fetal alcohol spectrum disorders. A total of 724 metabolites were detected and top 30 metabolites from maternal and fetal hepatic profiles were identified and analyzed. The experiment was solid and successful with sufficient data supported. Although limits to the distinguished unique metabolites, the result and conclusion are quite straight forward.

  1. What were the internal standards and / or database that this study applied for metabolites identification? If possible, please attach the related information in manuscript and/or supplementary information.
  2. What is the scree plot for the PCA analysis? And how about using more PCs for the development of models? If so, whether in the 3D-PCs plot, the metabolites could be distinguished?
  3. For the PLS-DA and oPLS-DA model, what were the S-plots (VIP>1) and volcano plot results? Would it be possible to pick up more significant changes metabolites (S-plot cutoff >0.5) that have potential hits the hepatic related pathway?
  4. If possible, what were the Q2 and R2 for the applied models? If possible, please attach the related information in manuscript and/or supplementary information.
